# Carbon Fibers Prepared via Solution Plasma-Generated Seeds

**DOI:** 10.3390/ma16030906

**Published:** 2023-01-17

**Authors:** Andres Eduardo Romero Valenzuela, Chayanaphat Chokradjaroen, Pongpol Choeichom, Xiaoyang Wang, Kyusung Kim, Nagahiro Saito

**Affiliations:** 1Department of Chemical Systems Engineering, Graduate School of Engineering, Nagoya University, Furo-cho, Chikusa-ku, Nagoya 464-8603, Japan; 2Department of International Collaborative Program in Sustainable Materials and Technology for Industries between Nagoya University and Chulalongkorn University, Graduate School of Engineering, Nagoya University, Furo-cho, Chikusa-ku, Nagoya 464-8603, Japan; 3Japan Science and Technology Agency (JST), Strategic International Collaborative Research Program (SICORP), Furo-cho, Chikusa-ku, Nagoya 464-8603, Japan; 4Japan Science and Technology Agency (JST), Open Innovation Platform with Enterprises, Research Institute and Academia (OPERA), Furo-cho, Chikusa-ku, Nagoya 464-8603, Japan; 5Conjoint Research Laboratory in Nagoya University, Shinshu University, Furo-cho, Chikusa-ku, Nagoya 464-8603, Japan

**Keywords:** carbon fibers, soot, solution plasma, solution plasma-generated seeds

## Abstract

Carbon fibers are materials with potential applications for CO_2_ capture due to their porous structure and high surface areas. Nevertheless, controlling their porosity at a microscale remains challenging. The solution plasma (SP) process provides a fast synthesis route for carbon materials when organic precursors are used. During the discharge and formation of carbon materials in solution, a soot product-denominated solution plasma-generated seeds (SPGS) is simultaneously produced at room temperature and atmospheric pressure. Here, we propose a preparation method for carbon fibers with different and distinctive morphologies. The control over the morphology is also demonstrated by the use of different formulations.

## 1. Introduction

Carbon materials have been proposed as promising materials for primordial applications, such as CO_2_ adsorption. Carbon materials have been used as adsorbents and for purification purposes throughout history. Porous solid materials, such as zeolites [1], metal–organic frameworks [2,3], covalent–organic frameworks [4,5], silicas [6], and activated carbons [7,8,9,10,11] have been broadly investigated owing to their advantages as adsorbents and potential use for CO_2_ capture. In particular, carbon materials have attracted attention for large-scale applications due to their chemical stability, low cost, ease of recovery, ease of handling, and cyclability. Another important feature of carbon materials is the different morphologies that they adopt. Fibrous morphologies, in particular, have major advantages over granular morphologies in terms of porous structure and larger surface areas. In terms of gas adsorption, micropores (<2 nm) play a major role. In granular porous carbons, gas needs to pass through macropores (>50 nm) and mesopores (2–50 nm) before reaching adsorption sites, whereas in fibers, most micropores are exposed directly to gas. As a consequence, the gas adsorption capacity of carbon fibers is higher than in granular carbons [9].

The most common methods for the preparation of carbon fibers are spinning and carbonization and vapor-grown [12,13,14,15,16,17,18,19]. In the case of the spinning method for achieving high-surface-area carbon fibers, precursors that give low crystallinity are recommended. Usually, pitch or polymers (e.g., PAN) are used. After spinning, there is a stabilization phase at 400 °C followed by a carbonization phase at 800–1200 °C for the preparation of these carbon fibers. For the vapor-grown fibers, the preparation follows a seeding catalyst approach in which catalyst particles are seeded on a substrate [14,15,20,21,22]. The factors for the growth of fibers are the precursor in gaseous a state, the substrate for the growth, and the atmospheric conditions (temperature, atmospheric composition, and pressure). This method has the advantage of simple and rapid preparation, a certain degree of crystallization in the final product, and synthesis of materials on a micronano scale. Nevertheless, some disadvantages should be pointed out, such as a lack of homogeneity in the radius and length of the fibers, and multiple preparation steps are required for the seeding of the catalyst.

Solution plasma (SP) is an emerging method for the synthesis of a variety of products, such as metal oxides, metal nanoparticles, and carbon materials with different structures [23,24,25,26]. By utilizing organic precursors, it is possible to synthesize carbon materials simply and continuously at room temperature and atmospheric pressure. A nonequilibrium system emerges due to the big temperature gradient between the center of the plasma and the surrounding liquid. In the SP system, there are two reaction pathways for the synthesis of carbon materials, one taking place at the center of the plasma and the other at the interface between the plasma and the liquid (plasma–liquid interface) [27]. The plasma–liquid interface hosts different reactions compared to the center of the plasma as seen in Figure 1. At the center, molecules are thermally degraded due to the high temperature (4000 K) and C2 species, and hydrogen and carbonized particles are released. At the plasma–liquid interface, there is a continuous flow of electrons in order to keep the plasma state, and secondary electrons are taken from the electrodes and liquid surroundings. When these secondary electrons are emitted, radical cations form in the solution. Radicals have a great reactivity and react with the surroundings via the radical polymerization process, and carbon materials are produced.

The advantages of this method are a fast synthesis rate of carbons, versatility in formulations, and process simplicity. The rapid synthesis of carbons is related to the molecular structure of the precursor. Ring molecules with π-delocalized electrons (e.g., benzene and dichlorobenzene) have a faster rate of carbon formation in the SP system. In contrast, linear molecules with σ bonds have a slower rate of carbon formation (e.g., NMP and ethanol). Nevertheless, controlling the morphology of carbon products in the SP system remains challenging due to the fast interaction of the reactants with the plasma. In the present study, we demonstrate the preparation of carbon fibers with a controlled morphology. During the discharge and formation of carbon in solution and disregarding the molecular structure of the precursor, a soot product-denominated solution plasma-generated seeds (SPGS) are simultaneously produced at room temperature and atmospheric pressure. Here, we explore the use of SPGS as a seeding material for the growth of carbon fibers. Different precursors, with different molecular structures, were employed, and their influence on the morphology of the final product was stablished.

## 2. Materials and Methods

### 2.1. Modified SP System

The modified SP system consisted of two parts, the SP reactor and the heating chamber. The SP reactor is a closed system comprised of a glass reactor and two tungsten electrode rods of 1.5 mm diameter (99.9% purity, Nilaco Corp, Tokyo, Japan) separated pin-to-pin by 1 mm. The electrodes with ceramic protection and two silicon stoppers were connected to a high-voltage bipolar generator (Pekuris AC-MV85-0002). A bubble diffuser was connected to the glass reactor for dispensing the carrier gas. Ar carrier gas (99.99% purity, Alpha System Co. Nagoya, Japan) was flowed at 30 sccm during the experiments. The second part comprised a gas delivery system and a plug flow reactor with a quartz tube and external heating source (modified furnace with a maximum temperature of 2000 °C). The stainless steel gas delivery system consisted of two valves, one for purging the air from the SP system before the plasma discharge and the other for allowing the SPGS to enter the plug flow reactor. The flow rate of the reaction gases (Ar 99.99% and H_2_ 99.99% Alpha System Co. Nagoya, Japan) was controlled by digital flowmeters.

### 2.2. Preparation of Nickel Foil Substrate

A Ni foil substrate (99.9% purity, Nilaco Corp, Tokyo, Japan) with a size of 30 × 150 × 0.3 mm was washed with 1 M HCl solution and rinsed with water and propanol. Then, the washed Ni foil was loaded inside the plug flow reactor under a continuous flow of Ar gas for drying any propanol remanent. The plug flow reactor remained sealed and air-free during the whole experiment. Next, the Ni foil was heat-treated at a temperature of 800 °C for 45 min to remove any organic matter or oxides from the surface. The flowing gases during heat treatment were Ar and H_2_ at 30 sccm. After heat treatment, the chamber was cooled to a reaction temperature of 400 °C and kept for 45 min before carbon fiber growth.

### 2.3. Synthesis of SPGS-Based Carbon Fibers Using Modified Solution Plasma (SP) System

Figure 2 shows the experimental setup for carbon fiber growth using SPGS from different liquid precursors. A total of 80 mL of cold benzene (99.0% purity), o-dichlorobenzene (99.0% purity) (hereafter referred to as dichlorobenzene), NMP (99.0% purity), ethanol (99.0% purity), or methanol (99.0% purity), all purchased from Kanto Chemical Co. Inc, Tokyo, Japan, were loaded in the SP reactor. All solutions were kept at 0 °C (7 °C for benzene) for 2 h before being loaded into the SP reactor. After loading the solution, the air was purged by Ar bubbling at 30 sccm. As soon as the plasma discharge started, the gas dispenser valve was opened to allow the SPGS to enter the plug flow reactor. SPGS was carried with Ar gas that flowed at 30 sccm. In order to keep the gas flow rate constant at 60 sccm, the Ar gas that flowed inside the plug flow reactor was stopped and exchanged with the SPGS carried by Ar gas (from the SP system). The plasma discharge was generated between the two electrodes using a high-voltage bipolar pulse generator (Pekuris AC-MV85-0002) using a voltage of 5 kV, frequency of 20 kHz, and pulse width of 1.0 µs. The plasma was kept constant for 15 min, and the solution temperature was monitored every 3 min, reaching a maximum of 40 ± 3 °C. After the reaction time, the plasma was stopped, and the H_2_ gas flow was stopped and exchanged for Ar at 60 sccm for cooling down to room temperature. Finally, the sample was collected, and the grown carbon fibers were sonicated from the substrate in acetone for 5 min.

### 2.4. Characterization

A field emission scanning electron microscope (FE-SEM, S-4800, Hitachi High-Technologies Co., Ltd., Tokyo, Japan) was used to observe the morphology of the carbon fibers with an acceleration voltage of 15 kV. The crystallinity was observed using X-ray diffraction equipment (XRD, Rigaku Co., Ltd., Smartlab, Tokyo, Japan) with monochromatic Cu Kα radiation (λ = 0.154 nm) operating at 45 kV and 200 mA. Raman analysis was measured by a spectrometer (Renishaw InVia TM Raman microscope) with a laser excitation wavelength of 532 nm. Specific surface area, pore volume, and average pore size of the carbon fibers were measured by the Brunauer–Emmett–Teller (BET) method, with N_2_ as adsorbate gas at liquid nitrogen temperature (77 K) using a BELSORP mini II analyzer (MicrotracBEL Corp, Nagoya, Japan) and analysis software BELMASTER TM v 5.3.3.0, provided by BEL Japan Inc., Nagoya, Japan.

## 3. Results and Discussion

### 3.1. Morphology of the Fibers

Figure 3 shows SEM images of various fibers prepared from different precursors grown on a Ni substrate at 400 °C. Figure 3a,b belong to fibers synthesized by benzene SPGS. Thick fibers, thin fibers, and amorphous carbon were formed simultaneously. The thickness of the fibers was between 150 and 220 nm, and their nonsmooth surface is characterized by the presence of bumps and grooves. Meanwhile, the surface of thinner fibers with diameters between 40 and 150 nm was quite smooth. In addition, Ni joints were observed as part of the fiber framework. Figure 3c,d present fibers grown by dichlorobenzene SPGS. Numerous fibers with diameters between 90 and 140 nm composed the bundle structure. Another remarkable characteristic is the metallic segmentation (brighter line perpendicular to the fiber axis) that indicates simultaneous columnar growth of the individual fibers. Worm-like fibers grown by NMP SPGS are shown in Figure 3e,f. This sample had diameters between 80 and 114 nm, and some fibers show a groove parallel to the fiber axis that might indicate irregular seed shapes and a turbulent growth. This growth type was considered a “fishbone” type of growth. Moreover, Ni particles were located at the tips of the fibers. Figure 3g,h exhibit entangled fibers with diameters of 38 nm and less fabricated by ethanol SPGS. Lastly, fibers grown by methanol SPGS formed “flower-like” clusters as in Figure 3i,j. Such clusters are comprised of thicker fibers of 72–76 nm in diameter and smaller and shorter fibers (around 20 nm in diameter). The particular shapes found and captured in SEM micrographs for different SPGS-grown fibers belong to repeatable findings using the same precursor. This means that fiber bundles, for example, grown by dichlorobenzene SPGS, cannot be found in samples using a different precursor. Nevertheless, an amorphous phase is present in all the samples.

### 3.2. Crystallinity of the Fibers

Figure 4a shows the XRD patterns of the synthesized carbon fibers grown by SPGS using different liquid precursors. The XRD patterns of all carbon fibers show a characteristic broad and sharp peak around 25.9° corresponding to the *002* plane of turbostratic graphite. The broadness of the peaks indicates the presence of small crystallites and amorphous structures. Ni forms part of the structure of the carbon fibers, and the sharp peaks around 44.5°, 51.9°, and 76.5° correspond to the *111*, *200,* and *220* crystallographic planes of Ni, respectively. In the case of carbon fibers grown by dichlorobenzene, the Ni peaks were sharper, which might indicate a significant metal content.

Figure 4b shows the Raman profiles of the synthesized carbon fibers. For all the carbon materials, the Raman results show two prominent peaks at around 1350 cm^−1^ and 1600 cm^−1^, corresponding to the well-defined D band and G band, respectively. In the case of carbon fibers grown by benzene, the intensity ratio of the D and G bands (I_D_/I_G_ ratio) < 1, and a sharper G peak indicates the presence of a certain crystallinity. For dichlorobenzene-grown fibers, the Raman profile showed an I_D_/I_G_ = 1, and the overlapping of the D and G peaks was proper for amorphous carbon. For the carbon fiber grown by NMP, the I_D_/I_G_ ratio was 0.8, including a sharp G peak, which indicated the presence of graphitic structures. Ethanol-grown fibers showed a sharper G peak and an I_D_/I_G_ ratio of 0.69, revealing the presence of a crystalline phase. Finally, methanol-grown fibers showed the I_D_/I_G_ > 1 and a sharper D peak similar to glassy carbons. In general, the profiles of the fibers for all the precursors indicate a predominant amorphous phase similar to black carbons with some turbostratic stacking showing certain crystallinity [9]. Moreover, the I_D_/I_G_ ratio can be used to estimate the defects of carbons, where a higher ratio ensures more defects in materials. The higher I_D_/I_G_ ratio depicts the defective nature of carbon material due to its porous and irregular structure [28].

### 3.3. Elemental Composition of the Fibers

The position of Ni in the carbon fiber structure was confirmed by energy-dispersive X-ray spectroscopy (EDX) in Figure 5. In the images, Ni was positioned at the tip and joint positions of the fiber structure. In general, Ni plays an important role in the growth of carbon fibers by providing a catalytic nucleus. Although the overall morphology was different depending on the precursor, all carbon fibers were formed according to the tip-growth mechanism by Ni as a catalyst [12,20]. As reported by Bartholomew C. in 1982 [29], in a temperature range between 200 and 500 °C, different carbon formations overlap on Ni, including carbides, amorphous carbon, and fibrous carbon. As seen in the micrographs in Figure 5, Ni was detected not only in the tips of the fibers but also in the centers of the fibers. The size of the Ni particles that were detached from the substrate also determines the diameter of the fiber, as can be seen in Appendix A.

### 3.4. Specific Surface Area of the Fibers

Figure 6 exhibits the N_2_ adsorption and desorption isotherm at the liquid nitrogen temperature of the carbon fibers prepared via SPGS. All fibers presented an isotherm type IV corresponding to mesoporous materials [30,31]. Moreover, the presence of a hysteresis loop represents the condensation of adsorbate in the mesoporous slit. Carbon fibers grown from benzene, dichlorobenzene, and NMP SPGS (Figure 6a,c) showed a hysteresis loop type H3, which corresponds to a mesoporous material with wedge slit-shaped pores. The specific surface areas were 188.02, 193.81, and 261.64 m^2^g^−1^ for benzene, dichlorobenzene, and NMP fibers, respectively. As shown in Figure 6d,e, fibers grown by ethanol and methanol showed hysteresis loops type H4 that correspond to parallel slit-shaped pores [31]. The specific surface areas of fibers grown by ethanol and methanol are 263.27 and 257.37 m^2^g^−1^, respectively. All the carbon fibers have a predominant mesoporous structure as seen in Table 1. Nevertheless, microporosity is present in NMP-grown fibers (around 10% of the total pore volume). Methanol-grown fibers present more microporosity than the other fibers (around 37% of the total pore volume). In contrast, ethanol-grown fibers, which showed a similar morphology than methanol-grown fibers, had a micropore volume of 0.019 cm^3^g^−1^.

### 3.5. Precursor Influences the Fiber Morphology

SP has the advantage of using different liquid precursors that would provide certain characteristics to the final product, mostly in terms of functionality. Nevertheless, morphology is difficult to control in this system. By using SPGS in combination with a plug flow reactor, it is possible to grow carbon fibers that have peculiar morphologies and textural properties. The precursor plays an important role in determining the morphological features of the SPGS-grown fibers owing to the different functional groups present in the molecular structure. Figure 7 shows the carbon fiber morphological peculiarities that depend on the precursor.

In the case of benzene SPGS-grown fibers, the precursor had a molecular structure characterized by the presence of π-delocalized electrons. This kind of molecule decomposed easily during SP. The reaction pathway followed reactions that produce pyrolytic and polymerized products (plasma center and plasma–liquid interface) [27]. Some of these molecules would have exited the SP system and joined the SPGS for the growth of fibers when they were carried by Ar and deposited on a Ni substrate. As seen in the SEM micrographs in Figure 3a,b, fibers with thicker diameters were found. The growth of fibers using benzene had been proposed by pioneer works [20,21] in which carbon fibers grew from catalyst seeds. In the present case, the Ni substrate worked as the growing catalyst media. Fibers nucleate from the SPGS precursor that agglomerated on the surface of the substrate where C_n_H_m_ species dehydrogenated by the cocatalyst action of the H_2_ atmosphere [32,33,34,35]. The agglomeration of seeds was favored in defects and boundaries found on the Ni substrate that acted as sinks. Moreover, Ni metal has been known for its affinity for absorbing carbon and promoting the growth of fibrous structures, planar structures, and graphitic and amorphous carbons [36]. The thickness of the benzene-grown fibers reflected that SPGS could agglomerate to a maximum that allowed the growth of 200 nm thick fibers, assuming that all native oxides and impurities were removed during pretreatment at 800 °C and only C_n_H_m_ species were present during growth at 400 °C. There were no agents that restricted the diffusion of seeds to a maximum fiber diameter (around 200 nm) using the proposed conditions for fiber growth.

The previous carbon fiber nucleation process differed when heteroatoms were introduced in the growing system. When using dichlorobenzene SPGS, carbon bundles nucleated from the metal surface (Figure 3c,d). Chlorine had the ability to corrode and soften the metal and promoted nanoparticle formation [37,38]. By using dichlorobenzene in the SP system, chlorine species were released and carried along SPGS. The surface of the Ni substrate was softened by these species. Agglomeration of carbon took place at groves and grain boundaries, where carbon fibers with diameters ranging 90–140 nm were grown simultaneously (columnar growth).

Contrary to π-delocalized molecules (benzene and dichlorobenzene), which have a fast synthesis rate of carbons in the SP system, σ-bonded molecules react slowly in the SP system. For these molecules, the reaction pathway for carbon formation was promoted by the center of the plasma, which decomposed molecules similarly to pyrolytic systems. In the case of NMP SPGS, carbon fiber growth was promoted by the decomposed species of this molecule containing carbonyl functional groups. NMP is known to have a corrosive effect on transition metals as it undergoes oxidation [39]. In the SP system, NMP molecules were decomposed in the center of the plasma, and both decomposed and vaporized species would join the SPGS for the growth of fibers. Oxygen also plays a role in the nucleation of deposited species, influencing the growth of graphene, for example [40]. In our system, NMP SPGS carried carbonyl groups that could have been responsible for the worm-like shaped fibers. When seeds agglomerated and were absorbed by the Ni substrate, an irregular growth of fibers was observed. The presence of a long groove along the axis of the fiber could be an indication of an irregular growth caused by the agglomeration of carbon around the Ni particle when the nucleation takes place. The presence of oxygen during the growth could have influenced such an irregular kind of growth. In general, the fibers seen in these samples resemble some of the benzene-grown fibers, except that their diameter and surface defects were different. These fibers also showed bigger surface areas due to surface defects and irregular shapes.

In the case of ethanol and methanol SPGS-grown fibers, these showed shorter and thinner fibers, with diameters < to 76 nm. Both ethanol and methanol interact similarly in the SP system. The carbon formation for both solvents was mediated by the reaction pathway at the center of the plasma. In addition, there were large amounts of solvent being evaporated. Both molecules carry hydroxyl groups that react at the surface of the substrate. These species also have softening properties [32,40,41]. These species dehydrogenate on the Ni substrate, and remaining oxygen species influence the growth of fibers. Moreover, as can be seen in Appendix A, hydroxyl functional groups also form part of the final product. The small size of the fibers seen for both ethanol and methanol fibers could also be related to the molecular size of the precursor that was evaporated. The clustering of entangled fibers happens due to the large amounts of SPGS produced during the SP process for both cases.

## 4. Conclusions

A new method for the preparation of carbon fibers, relying on the advantages of the solution plasma system, which provided a fast rate of carbon synthesis, easy setup, and operation at mild conditions, was developed.

The SPGS was synthesized at room temperature and atmospheric pressure and carried as the material for carbon fiber growth at 400 °C. Different molecular structures influenced the morphology of the final product. The factors influencing the fiber morphology were bonded to the nature of the precursor and how it interacted in the SP system. As a result, five different types of precursors were used, yielding different morphologies in all the cases. For the benzene SPGS-grown fibers, thick and bumpy fibers (200 nm in diameter) were characteristic. The collected material corresponded to mesoporous carbon with a surface area of 188.02 m^2^g^−1^ and a pore volume of 0.3097 cm^3^g^−1^. Dichlorobenzene SPGS-grown fibers showed fiber bundles comprising individual fibers with 90–140 nm in diameter. The collected material showed a surface area of 193.81 m^2^g^−1^, mesoporous nature, and a total pore volume of 0.4057 cm^3^g^−1^. Worm-like fibers (80–114 nm in diameter) grown by NMP SPGS showed a surface area of 261.64 m^2^g^−1^, pore volume of 0.3884, and mesopore dominance. Small fiber clusters (fibers having diameters < 78 nm) were characteristic of ethanol and methanol SPGS-grown fibers showing a predominant mesoporosity. The surface areas were 263.27 and 257.37 m^2^g^−1^, and the pore volumes were 0.2352 and 0.2139 cm^3^g^−1^ for ethanol and methanol SPGS-grown fibers, respectively. Overall, SPGS is a promising material for the preparation of rapid and morphologically controlled carbon fibers.

## Figures and Tables

**Figure 1 materials-16-00906-f001:**
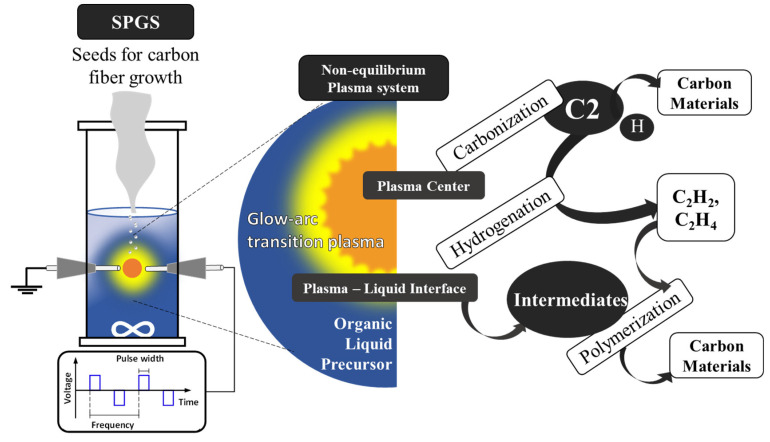
Concept for the utilization of SPGS for carbon fiber growth.

**Figure 2 materials-16-00906-f002:**
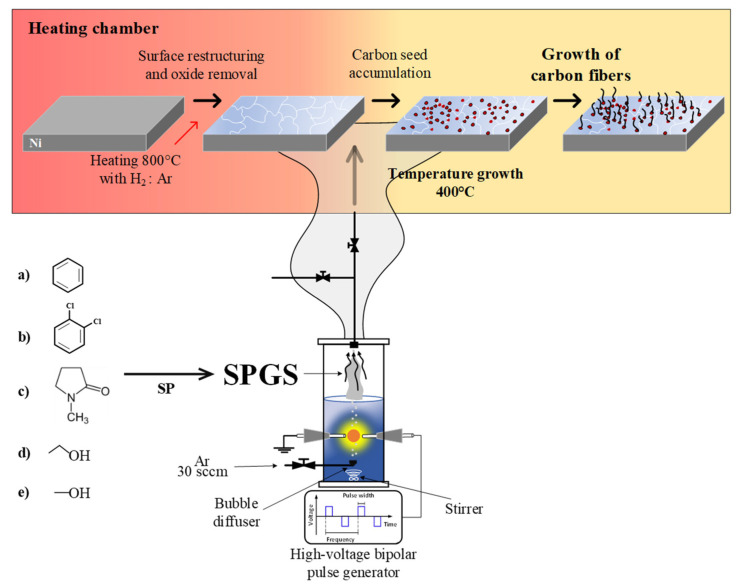
Experimental setup for the growth of carbon fibers using (a) benzene, (b) dichlorobenzene, (c) NMP, (d) ethanol and (e) methanol.

**Figure 3 materials-16-00906-f003:**
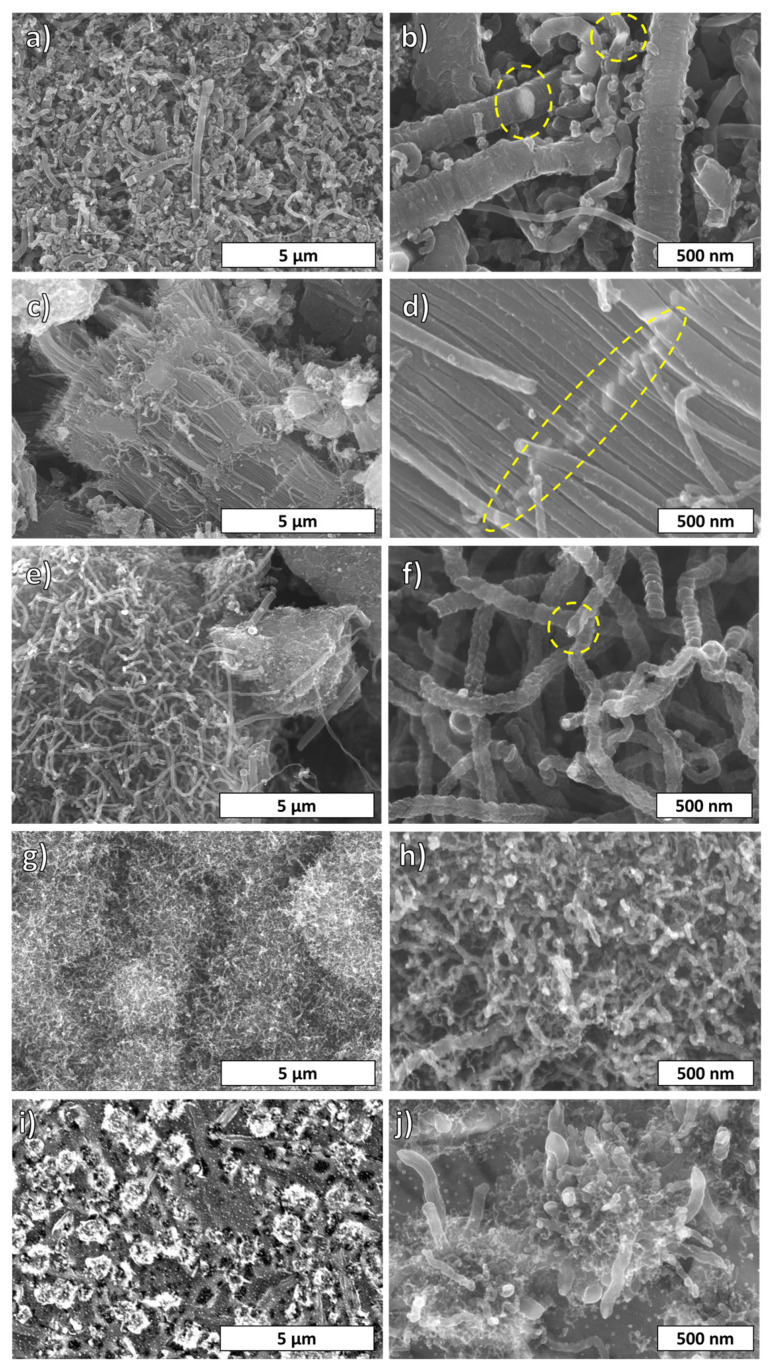
SEM micrographs of carbon fibers grown by (**a**,**b**) benzene, (**c**,**d**) dichlorobenzene, (**e**,**f**) NMP, (**g**,**h**) ethanol, and (**i**,**j**) methanol SPGS. Ni content at joints and tips is highlighted in yellow.

**Figure 4 materials-16-00906-f004:**
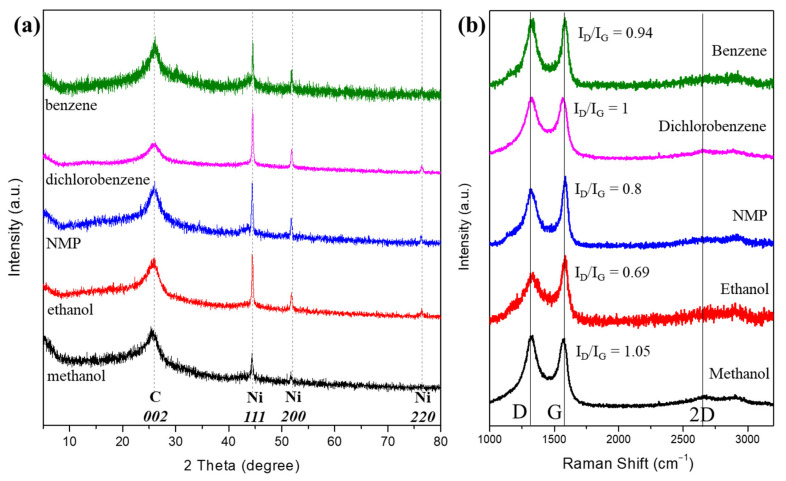
Crystalline structure of carbon fibers characterized by (**a**) XRD and (**b**) Raman spectroscopy.

**Figure 5 materials-16-00906-f005:**
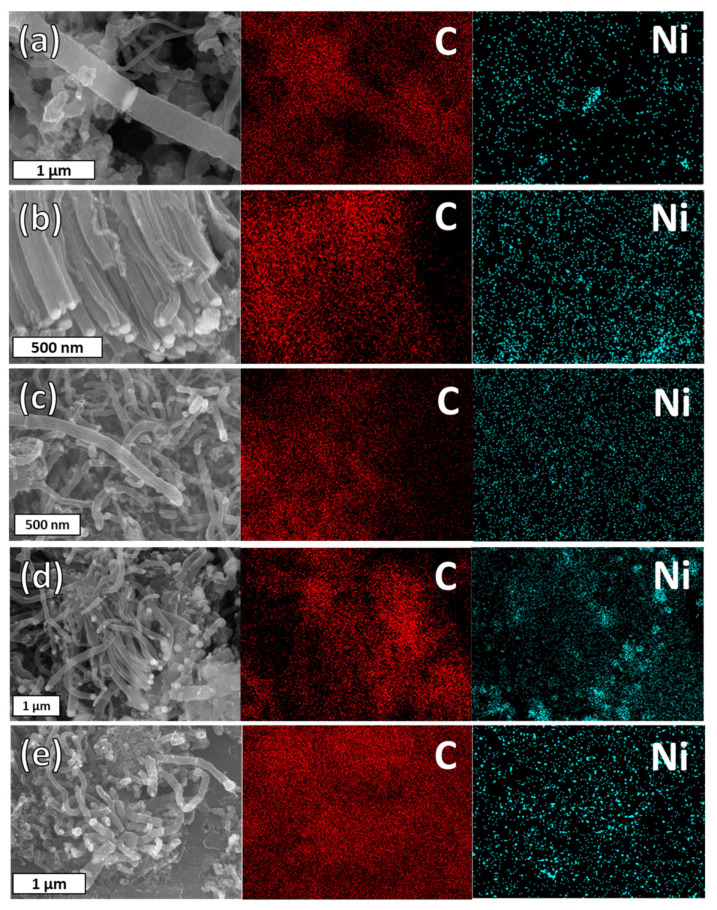
Elemental composition of carbon fibers grown by (**a**) benzene, (**b**) dichlorobenzene, (**c**) NMP, (**d**) ethanol, and (**e**) methanol SPGS by energy-dispersive X-ray spectroscopy (EDX).

**Figure 6 materials-16-00906-f006:**
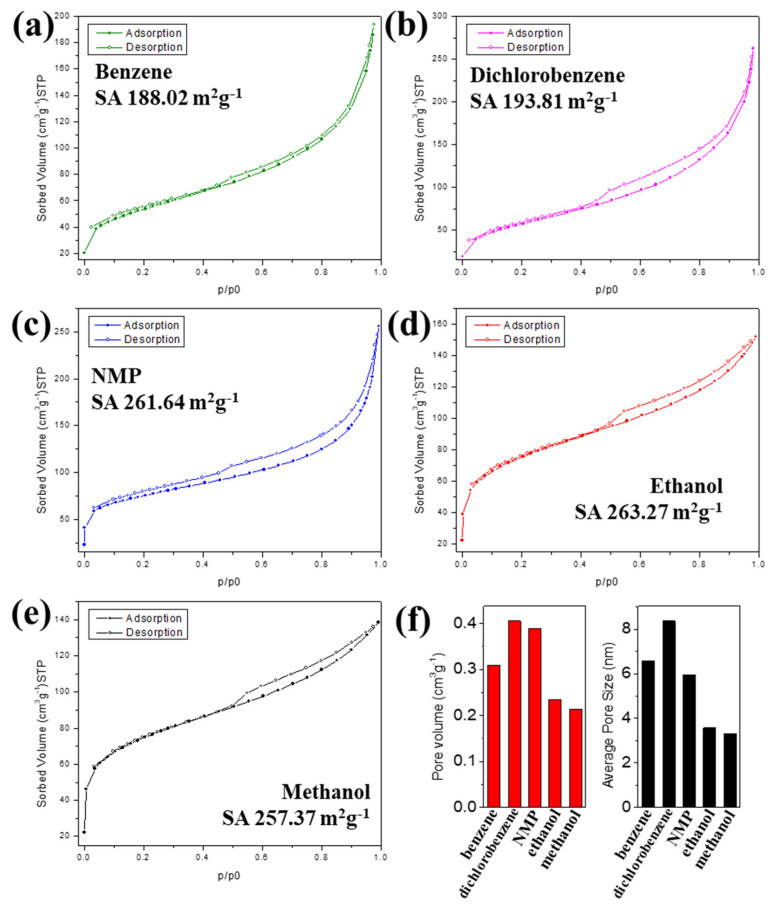
Nitrogen adsorption–desorption isotherms of carbon fibers grown by (**a**) benzene, (**b**) dichlorobenzene, (**c**) NMP, (**d**) ethanol, and (**e**) methanol SPGS at 77 K. (**f**) Pore volume and average pore size.

**Figure 7 materials-16-00906-f007:**
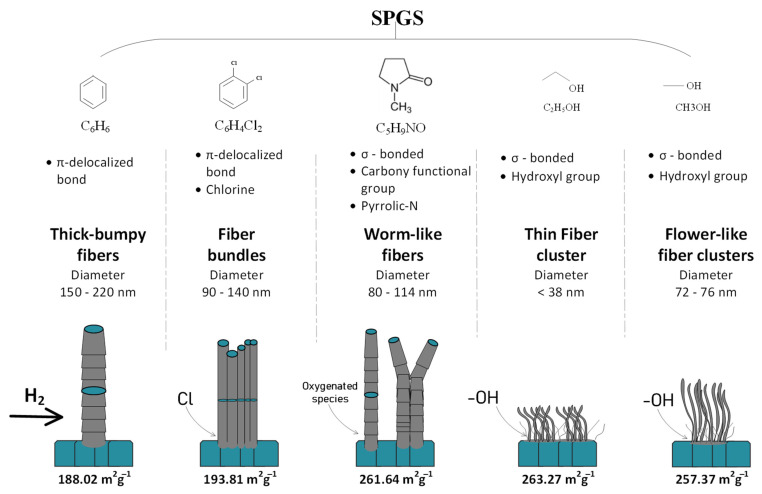
Precursor affecting the morphology of fibers.

**Table 1 materials-16-00906-t001:** Textural properties of carbon fibers grown by SPGS.

Sample	S_BET_ (m^2^g^−1^) ^(a)^	V_T_ (cm^3^g^−1^) ^(a)^	V_mic_ (cm^3^g^−1^) ^(b)^	V_meso_ (cm^3^g^−1^) ^(c)^	Average Pore Size (nm) ^(a)^
Benzene	188.02	0.3097	0.0083	0.3014	6.59
Dichlorobenzene	193.81	0.4057	0.0062	0.3995	8.37
NMP	261.64	0.3884	0.0406	0.3474	5.94
Ethanol	263.27	0.2352	0.0195	0.2157	3.57
Methanol	257.37	0.2139	0.0784	0.1355	3.32

^(a)^ Calculated by adsorption–desorption curves and BET method. ^(b)^ Micropore volume, calculated by t-plot method. ^(c)^ Mesopore volume V_meso_ = V_T_ − V_mic_. S_BET_: Brunauer–Emmett–Teller surface area. V_T_: Total pore volume. V_mic_: Micropore volume.

## Data Availability

Not applicable.

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
