# Peer review of "Carbon Fibers Prepared via Solution Plasma-Generated Seeds"

_materials, 2023, doi:10.3390/ma16030906_

Round 1

Reviewer 1 Report

The manuscript for review entitled "Carbon Fibers Prepared Using Seeds Generated by Solution Plasma" is written with great care and recommended for publication with minor suggestions:

- Figure 6f should be clearer (suggest to increase the caps on the axes)

- fig. 7 no superscripts next to the units

- it would be worth adding FTIR tests

- it is also worth reviewing the bibliography and replacing older references with newer ones

Reviewer 2 Report

Review of the Manuscript No materials-2148388

1. English text should be improved. Abstract, Introduction, Results and Conclusions should be rewritten.

Abstract, page 1, row 17-18

‘Carbon fibers are potential materials for CO2 capture owing to their porous structure and larger surface areas.’

Should be replaced by

‘Carbon fibers are materials with potential application for CO2 capture due to their porous structure and high surface area.’

2. Introduction should include the main aims of the work which are planned, and the tasks that will be performed.

3. Conclusions

‘Here we proposed a new method for the preparation of carbon fibers, relying on the advantages of the SP system which provides a fast rate of carbon synthesis, easy setup and operation at mild conditions.’

Should be corrected as follows

‘New method for the preparation of carbon fibers, relying on the advantages of the separation purification system which provides a fast rate of carbon synthesis, easy setup and operation at mild conditions, was developed’

4. Authors should correct the references according to journal requirements.

Reviewer 3 Report

In this work, the authors focused on the investigation of carbon fibers for CO2 capture, and presented a solution-plasma-generated seeds method for the preparation of carbon fibers. Distinctive morphologies were obtained by using different organic precursors. The authors also discussed the precursor influences on the fiber morphology. Overall, this work is well-organized, I recommend its publication in Materials after the following questions are well addressed.

1.     In the XRD patterns, the peaks around 43° using NMP and benzene are at the same position of peak for 111 plane of nickel with ethanol, how can it be attributed to 10l plane? And peaks for 222 and 220 planes of nickel in sample using ethanol are also found in the XRD patterns of samples using NMP and benzene precursors, so the peaks assignments should be the same.

2.     In line 213, the authors stated “The particle size Ni also determines the diameter of the fiber.”, how is this conclusion obtained?

3.     The values of surface areas for five samples are different in the main text, table 1, figure 6 and figure 7, please check carefully. 

4.     Some typos should be revised, eg. “2nm” in Line 38; period is missing in line 170; “1350 cm-1 and 1600 cm-1”in line 183; “ID/IG”in line 187 and 188; “a just a” in line 229; …

5.     The input methods are not consistent throughout the manuscript, please check carefully, eg. the caption of Figure 3.
